# Effect of Obesity on Aquaporin5 Expression in Human Placental and Uterus Tissues

**DOI:** 10.3390/jcm13154490

**Published:** 2024-07-31

**Authors:** Kata Kira Kemény, Zoltan Kozinszky, Ábel T. Altorjay, Bálint Kolcsár, Andrea Surányi, Eszter Ducza

**Affiliations:** 1Department of Pharmacodynamics and Biopharmacy, Faculty of Pharmacy, University of Szeged, H-6726 Szeged, Hungary; kemeny.kata.kira@szte.hu; 2Department of Obstetrics and Gynecology, Albert Szent-Györgyi Medical School, University of Szeged, H-6725 Szeged, Hungary; kozinszky@gmail.com (Z.K.); abel.altorjay@gmail.com (Á.T.A.); kolcsar.balint@med.u-szeged.hu (B.K.); gaspar-suranyi.andrea@med.u-szeged.hu (A.S.); 3Capio Specialized Center for Gynecology, Solna, 171 45 Stockholm, Sweden

**Keywords:** aquaporin, human pregnancy, obesity, uterus, placenta

## Abstract

**Background:** Obesity and overweight are also becoming more prevalent among women of childbearing age and pregnant women. In maternal obesity, the activation of metabolic, inflammatory, and oxidative stress pathways is proven, which appears to be a key step in the pathological changes observed in placental and uterine function. Several recent studies have evidenced that aquaporins (AQPs) are critical players in adipose tissue biology and are involved in the onset of obesity. **Methods:** Our studies aimed to investigate the changes in placental volume and vascularization and measure the AQP5 expression and total antioxidant capacity (TAC) in the placenta and uterus tissues in obese and typical-weight mothers. We also aim to measure the AQP5 plasma concentration. **Results:** We found AQP5 dominance in the uterus and plasma at 34 weeks of normal pregnancy. The placental volume increased and the vascularization decreased in obese mothers compared to the control. The AQP5 expression increased in the uterus of the obese group and did not change in the placenta. The TAC decreased in the plasma of overweight mothers. **Conclusions:** We hypothesize that increased AQP5 expression prolongs the length of pregnancy and inhibits the onset of contractions. Based on our findings, we can develop diagnostic tests and provide new targets for tocolytic drug development.

## 1. Introduction

Obesity is a multifaceted illness that has become a global epidemic. The global prevalence of obesity almost tripled between 1975 and 2023, primarily as a result of increasingly sedentary lifestyles and the consumption of unhealthy diets, with women being more affected than men [1]. A retrospective study presented population data in Hungary; Hungary’s obesity prevalence is higher than the regional average of 25.3% [2]. The prevalence rate of overweight and obesity among men was 40% and 32%, respectively, while among women, overweight and obesity occurred in 32% and 28% of cases [3]. In addition, obesity and overweight also affect women of childbearing age and pregnant women increasingly. Obesity is considered responsible for the increased risk of several diseases, such as type 2 diabetes, cardiovascular diseases, and musculoskeletal problems, and it is also related to malignancies or mental illness. The fact that maternal obesity leads to chronic inflammation in adipose tissue can contribute to inflammation by producing inflammatory mediators, which seems to be a crucial factor in the pathological changes in placental and uterine function [4]. Obesity exerts an independently increased risk of specific fetal malformations, namely neural tube defects, congenital cardiac anomalies, and orofacial clefts [5]. It is well known that obesity-induced oxidative stress represents the pathophysiological basis for most disorders, including reproductive issues. The above supports the study of factors that can change during obesity and affect pregnancy outcomes and fetal health.

Several recent studies have evidenced that aquaporins (AQPs) are critical players in adipose tissue biology and are involved in the onset of obesity [6]. The AQPs are a family of small (25–34 kDa) hydrophobic, integral membrane channel proteins that facilitate the rapid passive movement of water. AQP5 was found to be expressed in mouse adipose cells and play a crucial role in adipocyte differentiation [7], and AQP5-KO mice have a lower body weight than wild-type animals [8]. Interestingly, a high-fat diet induces the expression of AQP5 in subcutaneous adipose tissue [9], suggesting a role for AQP5 in adipose biology and cellular adaptation to fat accumulation. 

More AQPs were described in the mammalian female reproductive tract. In previous studies, we demonstrated the dominance of AQP5 during pregnancy in the rat uterus, with a dramatic drop on the last day of gestation. Aquaporin transport activity can be regulated by various factors, e.g., selectively regulated by oxytocin [10], and its expression is influenced by both estrogen and progesterone [11]. We have demonstrated the importance of AQP5 expression in pregnant rat uterus contractions and the initiation of preterm birth [12]. 

Amniotic fluid content is partially regulated throughout the placenta. This connection between the mother and the fetus is crucial for pregnancy. An inadequate amount of amniotic fluid is one of the indicators of abnormal placental function [13]. Volume abnormalities (polyhydramnios or oligohydramnios) increase fetal morbidity and mortality [14]. In human gestation, between weeks 10 and 14, chorionic villi showed high mRNA expression for AQP1, 3, 9, and 11, low levels for AQP4, 5, and 8, and no detectable AQP2, 6, and 7 [15].

Our studies determined the expression of the AQP5 isotype in the human uterus at different gestation ages. We also investigated AQP5 expression in human placental tissues at the end of pregnancy. Furthermore, we observed the impact of maternal obesity on the levels of AQP5 in the uterus and placenta and the antioxidant capacity of plasma.

## 2. Materials and Methods

### 2.1. Samples

A prospective, cross-sectional cohort study was conducted in pregnant women undergoing elective cesarean sections at the Department of Obstetrics and Gynecology, University of Szeged, Hungary, between June 2023 and March 2024. The inclusion criteria of the study were singleton pregnancies with an increased risk of fetal death because of intrauterine distress (fetal bradycardia and/or CTG anomaly and/or placental flow discrepancy), where a cesarean section was performed between 34 and 41 weeks of gestation. It should be noted that 34 + 0 and 40 + 0 weeks of gestation were recruited into our study. The study’s exclusion criteria were determined to include the following: multiple pregnancies, fetal or neonatal structural or genetic anomalies, improper localization of the placenta (such as placenta praevia), pathological placentation (placenta accreta spectrum), and self-reported drug, alcohol, or nicotine abuse.

The gestational age was determined using the first day of the last menstrual period and/or ultrasound biometry (crown–rump length and biparietal diameter) at the 10th week of pregnancy.

The uterus and placental samples (cc. 1 cm^3^) were collected at the cesarean section. The typical and obese women (BMI > 30 kg/m^2^) were carried out for our study at the Obstetrics and Gynecology Clinic, University of Szeged. The human study protocol was approved by the Clinical Research Ethics Committee of the University of Szeged (Ref. no. 57/2020-SZTE). This study was carried out according to the principles of the Declaration of Helsinki. We obtained written, informed consent from all participants. 

The human tissues were frozen immediately after collection and stored at −80 °C until assay. The blood samples were placed into polyethylene tubes containing EDTA and then centrifuged at 1500 g for 10 min at a temperature of 5 °C. The resulting plasma samples were then stored at −80 °C until they were ready for analysis. We could collect normal gestation (*n* = 10) uterus, placenta, and serum samples; obese gestation (*n* = 12) uterus samples; and obese gestation (*n* = 28) uterus with placenta and serum samples.

### 2.2. Ultrasound Investigations

The ultrasound investigation is performed 16–48 h before labor. Firstly, standard 2D measurements were taken: fetal position and presentation, body movements and fetal heart rate, and placental localization. The factorial default setting ‘Obstetrics/2–3 trimester’ was used in 2-D mode. 

Then, a 3D sweep was obtained through the placenta with power Doppler settings. The 3DPD sweep was obtained through the placenta with a Voluson S10 BT2021 ultrasound machine (RAB 2–5 MHz probe and 4D View version 10.4 program; GE Healthcare, Kretztechnik, Zipf, Austria). The angle of volume acquisition varied from 45 to 90° according to placental size. The volume acquisition was obtained in ‘maximum’ quality, and its duration was between 10 and 15 s, keeping the probe perpendicular to the placental plate. The multiplanar technique was applied, and after the entire volume was scanned, the three orthogonal ultrasonographic sections were analyzed and stored on a removable hard disk. The longest view of the placenta on the A plane of the three orthogonal ultrasonographic sections was chosen as the reference image. The same pre-established instrument settings were used in all the cases (power 96%; frequency low; quality normal; density 6, ensemble 16; balance 150; filter 2; smooth 3/5; pulse repetition frequency 0.9 kHz, gain −0.2). Each image was recovered from the disk in succession for processing. The stored volumes were further analyzed using the virtual organ computer-aided analysis (VOCAL) program pertaining to the computer software 4D VIEW (GE Healthcare Austria GmbH & Co OG. Tiefenbach 15 4871 Zipf Austria, version 10.4), which consists of outlining the contour of the placenta repeatedly after rotating its image six times by 30° with careful attention to exclude decidua and maternal vessels. After the complete rotation was finished, the placental volume was automatically calculated by the software. For each patient, placental volumes were measured three times by a specifically trained sonographer.

The three-dimensional volume is composed of small units of a volume known as “voxels”. Voxels encompass all data regarding gray and color intensity scales ranging from 0 to 100. Based on these values, three power Doppler indices can be derived from this measurement system, which are suitable for evaluating vessels and blood flow. These 3-DPD indices are employed in assessing placental perfusion, and it is widely acknowledged that they may reflect both uteroplacental and fetoplacental blood flow.

The vascularization index (VI), defined as the ratio of color voxels to total voxels, quantifies the presence of blood vessels within a given volume of interest and is expressed as a percentage (vascularity). The flow index (FI) represents the average color value of all color voxels, indicating the average intensity of blood flow on a scale from 0 to 100 (unitless). The vascularization flow index (VFI), calculated as the weighted ratio of color voxels to total voxels, combines information on vessel presence (vascularity) and the number of blood cells transported (unitless). The values range from 0 to 100. The VOCAL program automatically computes these indices (VI, FI, and VFI) based on the collected samples.

### 2.3. RT-PCR Studies



*Total RNA preparation from tissue*



The cellular RNA was isolated using the guanidinium thiocyanate–acid-phenol–chloroform extraction method described by Chomczynski and Sacchi [16]. Following precipitation with isopropanol, the RNA underwent a wash with 75% ethanol and was then resuspended in diethylpyrocarbonate-treated water. The purity of the RNA was verified by measuring the optical density at 260/280 nm using BioSpec Nano (Shimadzu, Japan), with all samples showing an absorbance ratio within the range of 1.6–2.0. The quality and integrity of the RNA were evaluated through agarose gel electrophoresis.



*Real-time quantitative reverse transcription-PCR (RT-PCR)*



Reverse transcription and amplification of the PCR products were performed using the TaqMan RNA-to-CT Step One Kit (Thermo Fisher Scientific, Hungary) and an ABI StepOne Real-Time cycler. Reverse-transcriptase PCR amplifications were performed as follows: at 48 °C for 15 min and at 95 °C for 10 min, followed by 40 cycles at 95 °C for 15 s and at 60 °C for 1 min. The generation of specific PCR products was confirmed by melting curve analysis. The following primers were used: assay ID Rn00562837_m1 for the Aqp5 water channel and Rn00667869_m1 for β-actin as an endogenous control (Thermo Fisher Scientific, Hungary). Each sample was analyzed in triplicate. The fluorescence intensities of the probes were graphed against the PCR cycle number. The amplification cycle at which the first significant increase in fluorescence signal was observed was designated as the threshold cycle (CT).

### 2.4. Western Blot Analysis

A total of 25 micrograms of protein per well were subjected to electrophoresis on 4–12% NuPAGE Bis-Tris Gel in XCell SureLock Mini-Cell Units (Thermo Fisher Scientific). Proteins were transferred from gels to nitrocellulose membranes using the iBlot Gel Transfer System (Thermo Fisher Scientific). Antibody binding was detected with the WesternBreeze Chromogenic Western Blot Immunodetection Kit (Thermo Fisher Scientific, Hungary). The blots were placed on a shaker and incubated with AQP5 (cat. no. AB-15858, 1:200, Sigma-Aldrich) and β-actin (cat. no. bs-0061R, 1:200, Bioss Antibody) polyclonal antibodies in the blocking buffer. Images were captured with the EDAS290 imaging system (Kodak Ltd., Hungary), and the optical density of each immunoreactive band was determined with Kodak 1D Images analysis software (IS2000R). Optical densities were computed as arbitrary units following the subtraction of the local area background.

### 2.5. Total Antioxidant Capacity and ELISA Assay

Following the manufacturers’ recommendations, plasma total antioxidant capacity (T-AOC) and AQP5 concentration were measured using a colorimetric assay kit (T-AOC, Elabscience, USA) and an ELISA assay kit (AQP5, FineTest, China). All optical density values were measured using a SPECTROStar Nano microplate spectrophotometer (BMG Labtech, Germany). 

### 2.6. Statistical Analysis

Statistical analyses were conducted using Prism 10.2.1 software (Graphpad Software Inc., San Diego, CA, USA). The data were assessed using either a one-way ANOVA test with Dunnett’s post hoc test or an unpaired *t*-test, and results are reported as the mean ± SEM. Statistical significance was considered at *p* < 0.05.

## 3. Results

### 3.1. Placental Sonography Studies

Descriptive statistics are shown in Table 1. In our obese group, the mean maternal age (34.52 years) was slightly higher compared to the national average reference age at delivery, which was 29.24 years for primiparous women and 30.54 years for the total number of pregnant women in 2023 [17]. Obese women who participated in our study had a mean BMI of 36.17 kg/m^2^ at the first medical visit. In the normal group, 32% and 24% of the obese group were primiparous. The gestation age at delivery was not significantly different, but the fetal weights in the obese group were significantly higher. 

Figure 1 presents a placental volume analysis by the virtual organ computer-aided analysis (VOCAL) program pertaining to the computer software 4-D VIEW (GE Medical Systems, Austria, version 10.4). The most accurate volume analyses were prepared by three-dimensional ultrasound. Each image was recovered from the disk in succession for processing. The stored volumes, marked by outlining the contour of the placenta repeatedly after rotating its image six times by 30° with careful attention to exclude decidua and maternal blood vessels, were analyzed using the VOCAL. After the complete rotation was finished, the software automatically calculated the placental volume. The intraobserver correlation coefficient (0.97) regarding measuring placental parameters was excellent.

Figure 2 displays the 3-dimensional power Doppler indices in the placenta. The in vivo vascular analyses were at umbilical insertion in the most vascularized part of the placenta. In the two study groups, the placental volume was significantly elevated, while the vascularization indices (VI: vascularization index, FI: flow index, and VFI: vascularization flow index) were significantly depressed in the obese cases compared to typical-weighted pregnant women (Table 2). 

### 3.2. Molecular Biology Studies

The AQP5 mRNA expression (Figure 3A) in the uterus did not change significantly from the 36th to the 41st weeks of a normal pregnancy. Only protein expression was reduced at week 36 of gestation (Figure 3B). Surprisingly, the AQP5 mRNA and protein expression showed a dramatic decrease in the uterus at the 36th week of pregnancy compared to the 34th week (Figure 3).

The AQP5 concentration significantly increased in the plasma samples of pregnancy weeks 34 (Figure 4) compared to the non-pregnant control and other weeks of normal pregnancy.

The AQP5 mRNA (Figure 5A) and protein expression (Figure 5B) significantly increased in the uterus of overweight mothers compared to mothers of typical weight.

No changes in AQP5 mRNA (Figure 6A) and protein (Figure 6B) expression were found in the placental tissue of the obese mothers compared to mothers with typical body weight.

We measured the total antioxidant capacity (TAC) in the plasma of obese and typical-weight mothers. In overweight women, the TAC significantly decreased compared to the control pregnant woman (Figure 7).

## 4. Discussion

Obesity is a chronic systemic disease that impacts all tissues and organs within the body, leading to an increased risk of various common conditions. The reproductive system is not exempt from the adverse effects of obesity. Obesity is typically categorized using the body mass index (BMI), which is calculated by dividing weight in kilograms by height in meters squared (kg/m^2^). According to the World Health Organization (WHO), obesity is defined as a BMI greater than 30 kg/m^2^, and overweight is defined as a BMI greater than 25 kg/m^2^ [18]. 

In a sonographic-based cohort study of singleton pregnancies, the estimated placental volumes and fetal weights are higher in pregnancies with obesity compared to the typical-weighted controls. Our data may be consistent with the report that obese mothers have larger placentas, their offspring’s weight is higher, and there is an increased placental ratio, which refers to the ratio of placental weight to fetal weight. Our study suggests disproportionate fetal growth compared to placental development in obesity occurs in late pregnancy.

A negative correlation was observed between 3-D power Doppler indices and maternal BMI in pregnancies with obesity. The control group and the obese groups had overlaps of VI, FI, and VFI values, but all placental vascular indices showed a tendency for a slight reduction in cases complicated by obesity. However, the increased volume expansion of the placenta was observed in connection with BMI. Decreased placental VI and VFI in obesity could be associated with down-regulated angiogenesis and a diminished number of arterioles. At the same time, the depressed FI might represent a narrower inner vascular diameter. 

During pregnancy, obese women are at an increased risk of early and late miscarriage, as described in the literature [19]. This risk is observed in both natural and assisted conceptions and is shown to increase in a manner dependent on body mass index (BMI) [20]. An additional issue associated with obesity during pregnancy is the increased incidence of cesarean sections. The primary reason for the elevated rate of cesarean sections in obese women, as opposed to non-obese women, is attributed to an altered myometrial function that results in decreased frequency and effectiveness of contractions. Our previous studies have found a negative correlation between AQP5 expression and uterine contraction, suggesting that a decrease in AQP5 expression results in increased myometrial activity and induction of labor in the rat uterus [12]. Therefore, we considered it essential to determine the ontogeny of AQP5 expression between 34 and 41 weeks of human pregnancy. We measured high AQP5 levels at week 34, which decreased significantly by week 36 and remained low until the end of pregnancy and the last week of the study. We hypothesize that this may support a role for the decreasing expression of AQP5 in the regulation of uterine contractions, in agreement with the results obtained in animal studies. Our hypothesis is supported by the fact that AQPs are also involved in the contraction of other smooth muscle-maintaining organs, such as airway smooth muscle. C.M. Krane and his colleagues utilized Aqp5 knockout (Aqp5−/−) mice in order to examine the role of AQP5 in lung physiology. In comparison to mice with the Aqp5+/+ genotype, mice lacking the *Aqp5* gene (Aqp5−/−) demonstrate a markedly increased concentration-dependent bronchoconstriction in response to intravenously administered acetylcholine [21]. This examination also proved the importance of AQP5 in smooth muscle contraction.

We also examined plasma AQP5 concentrations to determine its physiological changes and may use these data as a diagnostic marker for future identification of pathological lesions. The change in plasma and uterine AQP5 expression was correlated with a significant increase in week 34 and was significantly low and unchanged in the other weeks studied. We suspect that several factors may have influenced this. There is widely recognized evidence demonstrating the advantageous impact of antenatal corticosteroids on fetal lung maturation, leading to widespread recommendations for the utilization of this treatment in women who are at risk of preterm delivery. Women who took part in our studies and delivered at gestational week 34 were also treated with betamethasone. We may suppose that glucocorticoid therapy influenced AQP5 expression. Chen-Jie Yu et al. determined that glucocorticoids induce AQP5 expression in the sinonasal mucosa of chronic rhinosinusitis rats [22]. Moreover, cortisol modulates the water permeability of AQP2 via the rapid non-genomic effects of corticosteroids [23]. Corticosteroids have been shown to increase the expression of Aqp5 mRNA and AQP5 protein in human alveolar basal epithelial cells, suggesting a potential improvement in pulmonary function for preterm neonates. *AQP5* gene promoter methylation may be cell- and tissue-specific (e.g., lung). High methylation of the *AQP5* gene promoter was associated with low AQP5 expression in alveolar epithelial cells. In vitro, methylation of the *AQP5* gene promoter inhibited the transcription of a reporter gene in mouse lung epithelial cells, which supports our hypothesis that AQP5 reduction is significant in response to corticosteroid treatment [24].

On the other hand, changes in AQP5 can also provide information about the placenta’s status. By week 12, the placenta is formed and ready to provide nourishment for the baby. It is considered mature after 34 weeks. Villous maturation typically occurs several weeks prior to or near term, seldom before 34 weeks of gestation, and is commonly linked to maternal metabolic disorders and obesity.

The main obstetric risk factors for preterm birth are preeclampsia, premature rupture of the membranes, rupture of the placenta, and idiopathic cases. For preterm births, the 34th week of pregnancy is also therapeutically significant. Vaginal progesterone demonstrated a significant reduction in the risk of preterm birth or fetal death by 34% in women no older than 34 weeks of gestation [25]. Based on this, we suppose the diagnostic value of the changes in AQP5 expression in the plasma for preterm birth, but this requires further investigation. 

The onset of parturition in obese women is often postponed. In the absence of induction, obese women are nearly twice as likely as women of typical weight to experience prolonged pregnancy (≥41 weeks gestation), especially those with a BMI of 35 kg/m^2^ or higher [26]. Maternal obesity was associated with higher doses of used prostaglandins, the less frequent success of cervical ripening methods, and higher doses of synthetic oxytocin, as well as a longer time to birth after oxytocin use. AQP5 expression was significantly higher in the uterus of obese women compared to the control mother group. In contrast, we did not find changes in AQP5 expression in the placental tissues. Our group has more results from animal studies about the role of AQP5 in uterus contraction. The dropping of AQP5 expression was proved to be an inducing factor of uterus contraction and premature labor in our earlier studies. Based on these results, we may have assumed that the high concentration of AQP5 in the late pregnant uterus can delay the time of birth. It can be assumed that the AQP5 and other factors, e.g., the oxytocin and progesterone [10,11] combination effect, affect the length of pregnancy and the outcome of childbirth. 

Reactive oxygen species (ROS) and free radicals within a cell can harm essential compounds such as lipids, proteins, DNA, carbohydrates, and enzymes, ultimately causing cellular damage. Pregnancy itself is considered an oxidative event, yet in normal pregnancies, there appears to be a delicate balance between antioxidant and oxidant levels, despite some level of oxidative stress. Research has shown that an imbalance in oxidant/antioxidant status may play a role in the development of various obstetrical complications. As the gestational weeks progress, the oxidant–antioxidant equilibrium fluctuates, with oxidation processes intensifying throughout pregnancy. Particularly in the later stages of pregnancy, there is a notable increase in free radicals, prompting a corresponding rise in antioxidant mechanisms to counteract the heightened oxidative stress [27]. Our results show that this vital antioxidant capacity is decreased in obese mothers, which can lead to perinatal and postpartum complications. These results correlated with more studies that have suggested that systemic oxidative stress correlates with BMI [28] and that increased oxidative stress in accumulated fat is, at least in part, the underlying cause of the lack of regulation of adipocytokines and the development of metabolic syndrome [29].

## 5. Conclusions

Our results suggest that obesity elevates placental volume and depresses vascularization indices. AQP5 expression increases in the obese uterus but does not change in the obese placenta compared to a typical body weight in the third trimester. Moreover, overweight induces a decrease in the total antioxidant capacity, which can be the cause of obesity-related diseases. With our experiments, we gained information about the possible correlation between animal and human studies on AQP5 expression and uterus contraction. Furthermore, our research may answer important pathological processes induced by obesity during pregnancy. These results can form the basis for developing diagnostic tests and provide new targets for tocolytic drug development.

## Figures and Tables

**Figure 1 jcm-13-04490-f001:**
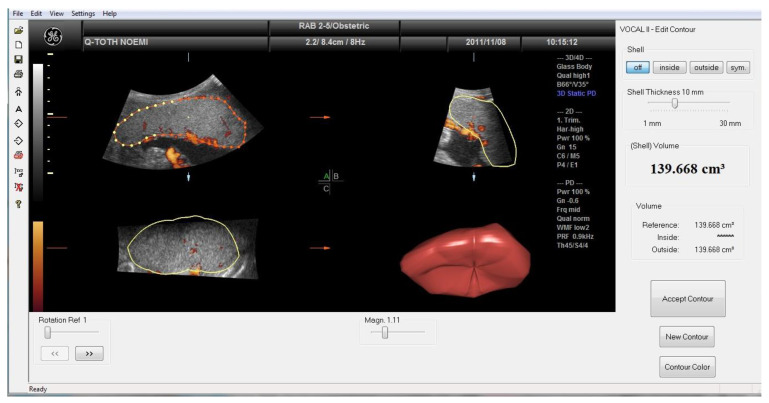
Determination of placental volume by VOCAL program. The yellow dotted lines represent the border of the placenta on reference plane ‘A’. The yellow lines are mapped from plane ‘A’ to represent the border of placenta on plane ‘B’ and ‘C’.

**Figure 2 jcm-13-04490-f002:**
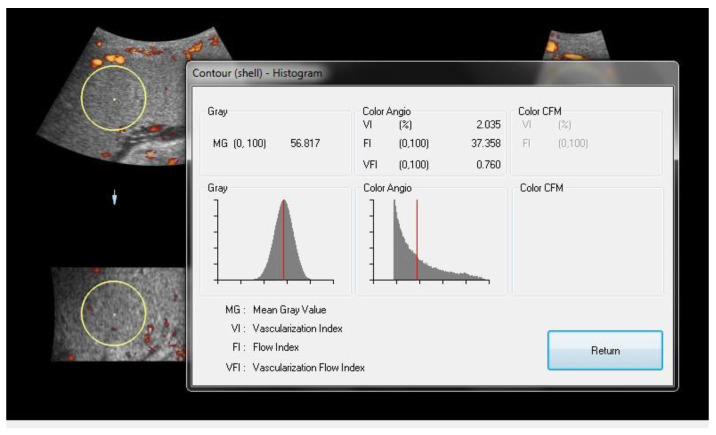
The VOCAL program shows the results of the histogram analysis of the sphere volume obtained automatically on the screen. The color angio analysis presented VI (vascularization index), FI (flow index), and VFI (vascularization flow index) values. The yellow circle represents the sonobiopsy area.

**Figure 3 jcm-13-04490-f003:**
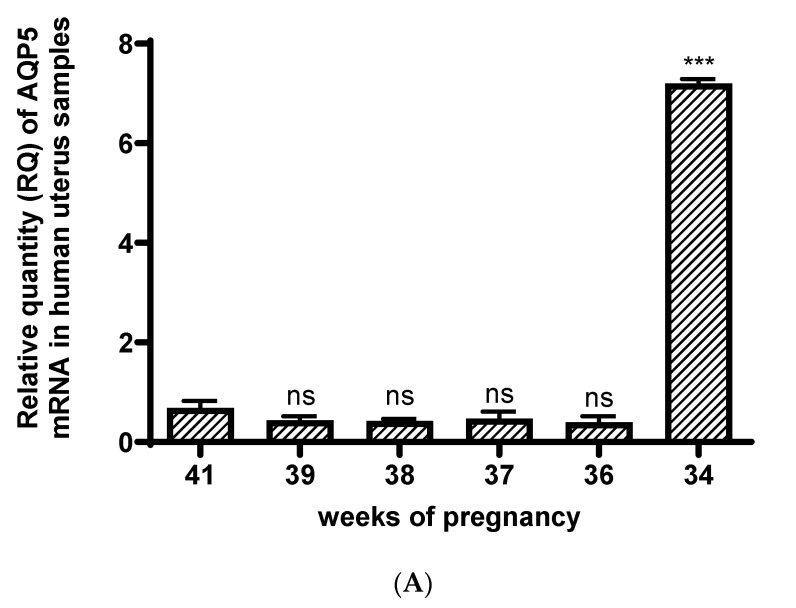
Changes in the AQP5mRNA (**A**) and protein (**B**) expression during different weeks of normal pregnancy in human uterine samples; ns: *p* > 0.05, * *p* < 0.05, *** *p* < 0.001 compared to the 41st week of pregnancy (The western blot pictures are attached as a Appendix A).

**Figure 4 jcm-13-04490-f004:**
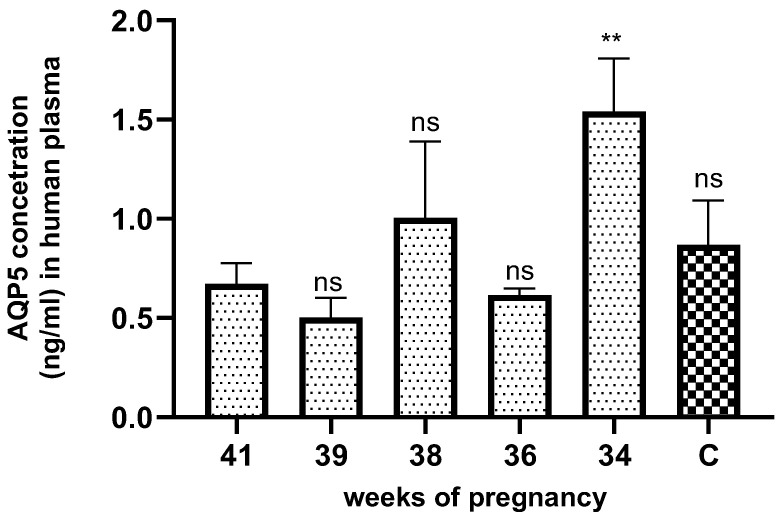
AQP5 concentration in the different weeks of normal pregnant and non-pregnant (C) human plasma samples. ns: *p* > 0.05, ** *p* < 0.01 compared to the control.

**Figure 5 jcm-13-04490-f005:**
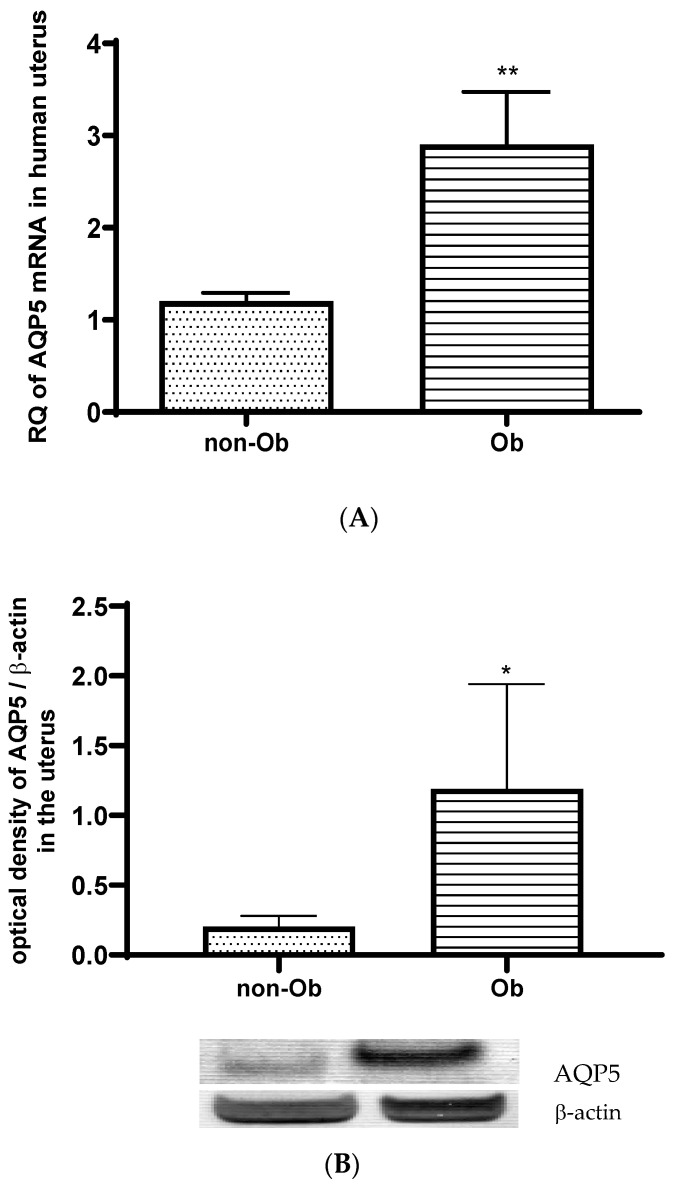
Changes in AQP5 mRNA (**A**) and protein (**B**) expression in uterine samples with typical weight (non-Ob) and obese (Ob) mothers in the third trimester of pregnancy; * *p* < 0.05, ** *p* < 0.01 compared to the control. (The western blot pictures are attached as a Appendix A).

**Figure 6 jcm-13-04490-f006:**
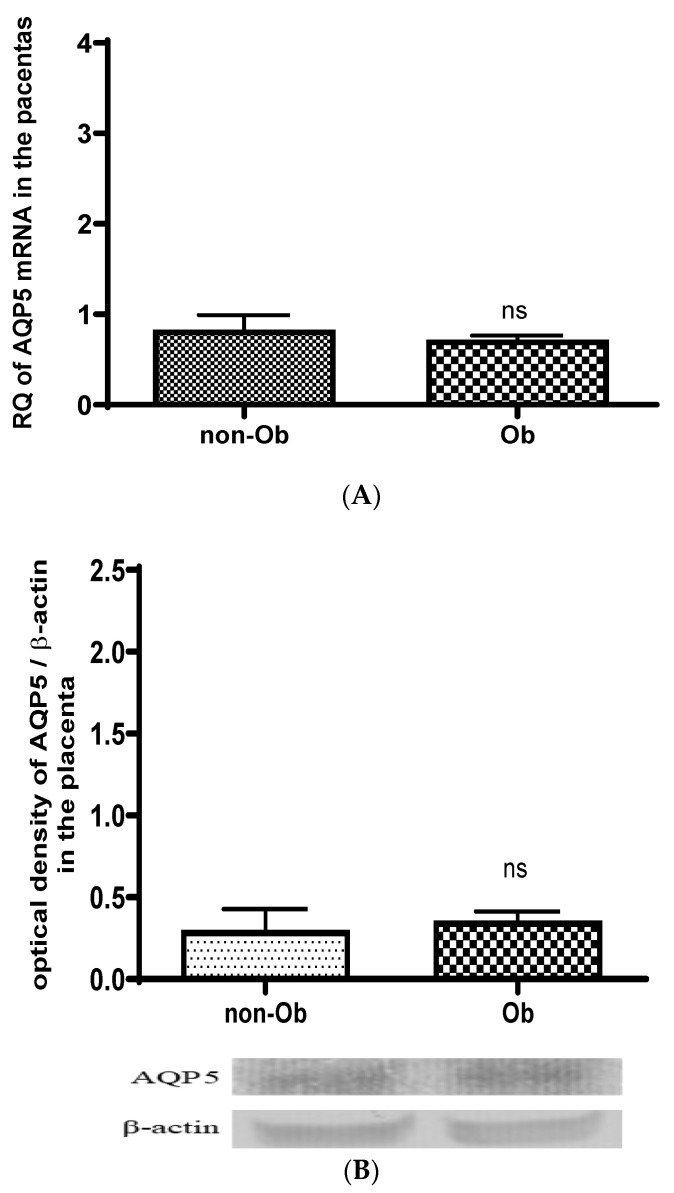
Changes in the AQP5 mRNA (**A**) and protein (**B**) expression in the placental tissue samples collected from obese (Ob) and typical-weight mothers (non-Ob) in the third trimester of pregnancy; ns *p* > 0.05 compared to the non-obese mothers (The western blot pictures are attached as a Appendix A).

**Figure 7 jcm-13-04490-f007:**
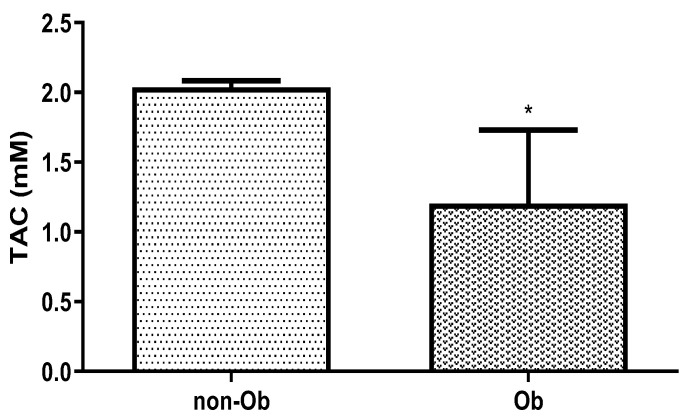
Change in total antioxidant capacity (TAC) in human plasma in typical weight (non-Ob) and obese (Ob) mothers in the third trimester of pregnancy. * *p* < 0.05, compared to the control (non-Ob) human plasma.

**Table 1 jcm-13-04490-t001:** Demographic characteristics of the groups typical-weighted and obese pregnant women. Statistically significant differences are indicated by *** for *p* < 0.001, while ns denotes no significant difference (*p* > 0.05) compared to the normal body weight group.

Demographical Data	Normal	Obese
Maternal age (years) (mean ± SD)	29.44 ± 1.1	34.52 ± 5.78
Number of nulliparous women in the study (%)	32	24
BMI at the time of first medical visit (kg/m^2^)	22.98 ± 2.90	36.17 ± 5.1 ***
Gestational age at delivery (weeks)	39.05 ± 1.2	39.25 ± 2.3 ^ns^
Fetal weight at delivery (grams)	3224.08 ± 243.54	3824.08 ± 538.63 ***

**Table 2 jcm-13-04490-t002:** The results of placental sonography.

Placental Sonography	Normal	Obese
Placental volume (mL ± SD)	527.3 ± 93.1	775.6 ± 143.2 ***
VI (mean ± SD)	14.11 ± 5.1	8.71 ± 2.4 ***
FI (mean ± SD)	44.97 ± 22.64	37.4 ± 10.9 *
VFI (mean ± SD)	8.21 ± 3.63	4.74 ± 1.34 *

VI: vascularisation index, FI: flow index, and VFI: vascularization flow index. * *p* < 0.05, *** *p* < 0.001 compared to pregnant women with normal body weight.

## Data Availability

The data can be accessed by email from the authors.

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
