# Peer review of "Effect of Obesity on Aquaporin5 Expression in Human Placental and Uterus Tissues"

_jcm, 2024, doi:10.3390/jcm13154490_

Round 1

Reviewer 1 Report

Comments and Suggestions for Authors

Dear Authors,

Thanks for this interesting paper. Your study, that seems in the area of your interest and expertise, has a scientific impact regarding obesity and pregnancy, especially in the field of aquaporine. The study design is very complete and complex, I appreciated the tables (very clear to read) and image, especially the use of US 3D for placenta. The introduction provided is very clear and satisfactory, consistent with an up to date discussion. My main concern regards the section of material and methods: why did you put it after results and discussion? I would appreciate, as always, the methods section before the result section. Moreover, due to its complexity, I would you to clarify the methods and materials: some sentences, especially the inclusion criteria and the included women, are not clear and by mistake not completed. Please clarify and move the methods section, paying particular attention to the inclusion criteria. You cited uterus samples, what do you mean? Did you analyze biopsies, if yes, how you get them? Or did you analyze full uterus? US? Histopathology? Please be more clear also regarding the gestational age of the included women. Furthermore, you cite 'high risk of fetal death' as included criteria. Please clarify what you mean with this, defining this criteria.

For the nature of the paper and its complexity, I would appreciate a conclusion, which is now missing. Please follow introduction, materials and methods, results, discussion, conclusion.

Best regards   

Comments on the Quality of English Language

The use of the language is appropriate. Some sentences in the methods sections are incorrect by mistake, I addressed my suggestions. 

Author Response

MS no: jcm-3116945

The authors would like to thank Reviewer #1 for the questions, which have promoted the creation of a better manuscript concerning its scientific value. Our answers are given below. Changes are marked with yellow in the manuscript.

Comment 1- My main concern regards the section of material and methods: why did you put it after results and discussion? I would appreciate, as always, the methods section before the result section.

Response 1- We changed the position of Material and methods, as the reviewer suggested.

Comment 2- Moreover, due to its complexity, I would you to clarify the methods and materials: some sentences, especially the inclusion criteria and the included women, are not clear and by mistake not completed. Please clarify and move the methods section, paying particular attention to the inclusion criteria.

Response2 - Line 79-83: Inclusion criteria of the study were singleton pregnancies with increased risk of fetal death because of intrauterine distress (fetal bradycardia and/or CTG anomaly and/or placental flow discrepancy), where a cesarean section was performed between 34-41 weeks of gestation. The 34+0 and 40+0 weeks of gestation were recruited into our study.

Comment 3- You cited uterus samples, what do you mean? Did you analyze biopsies, if yes, how you get them? Or did you analyze full uterus? US? Histopathology?

Response 3- The uterus and placental samples (cc. 1 cm3) were collected at the cesarean section. Normal and obese women (BMI > 30 kg/m2) were carried out to our study at the Obstetrics and Gynecology Clinic, University of Szeged.

Comment 4- Please be more clear also regarding the gestational age of the included women.

Response 4- Line 82. The 34+0 and 40+0 weeks of gestation were recruited into our study.

Comment 5- Furthermore, you cite 'high risk of fetal death' as included criteria. Please clarify what you mean with this, defining this criteria.

Response 5- Line 81-82 These factors were fetal bradycardia and/or CTG anomaly and/or placental flow discrepancy.

Comment 6- For the nature of the paper and its complexity, I would appreciate a conclusion, which is now missing. Please follow introduction, materials and methods, results, discussion, conclusion.

Response 6- We completed the manuscript with the conclusion part.

Reviewer 2 Report

Comments and Suggestions for Authors

The article is devoted to the study of the role of AQP5 during pregnancy in the 3rd trimester and the state of the total antioxidant capacity in women with normal weight and obesity. The study of the peculiarities of fetal-maternal interaction during pregnancy in obese women is an important task because maternal overweight affects placental function and relates to risk factors for fetal malformation. Finding the factors that may change by obesity and affect pregnancy outcomes and fetal health is essential for developing diagnostic tests and therapeutic methods. This clinical study was based on the results obtained by the authors in earlier experimental work on laboratory animals, and confirmed the important role of AQP5 in mother-placenta-fetal interactions and the effect of obesity on these interactions.

Undoubtedly, the results of this work are relevant and contain new information about the role of AQP5 in the third trimester of pregnancy in women with normal weight and obesity. The results may be useful for developing diagnostic and treatment methods for complications of pregnancy.

There are some questions for the authors:

1.   The authors did not specify the number of women in each study group and the distribution type of values, so it is difficult to assess the validity of the use of statistical methods and the presentation of results. Authors should indicate the number of observations in groups.

2.       The authors point to a dramatic increase of the AQP5 mRNA and protein expression in the uterus at the 34th week of pregnancy, but they have not examined women in earlier periods. In this case, we can talk about a dramatic decrease of 36 weeks. It is not clear for which study group the results are presented in Fig.3. A similar observation should be made for AQP5 concentration in blood plasma (Fig.4). Is there any information available for the authors that reveals AQP5 levels in the uterus and blood during the second trimester?

3.        Fig.5-7 compares AQP5 and TAC levels in women with normal weight and obesity, but the authors do not indicate when the study was conducted. The authors should indicate the age of pregnancy for these indicators.

4.       The authors write that their data may be consistent with the report that obese mothers have larger placentas, than normal weight woman. It is important to include a reference to the report.

5.       According to the authors, placental volumes are higher in pregnant women with obesity than normal weights, but all placental vascular indices show a slight decrease in cases complicated by obesity. Could it be due to increased water and placental edema?

Author Response

MS no: jcm-3116945

Answers to Reviewer #2

The authors would like to thank Reviewer #2 for the questions that have promoted the creation of a better manuscript concerning its scientific value. Our answers are given below. Changes are marked with yellow in the manuscript.

  1. The authors did not specify the number of women in each study group and the distribution type of values, so it is difficult to assess the validity of the use of statistical methods and the presentation of results. Authors should indicate the number of observations in groups.

Response 1:  We completed the manuscript with the number of cases in lines 100-102.

Normal gestation: n=10 uterus, placenta, and serum samples; obese gestation: n=12 uterus samples; and obese gestation: n=28 uterus with placenta and serum samples.

  1. The authors point to a dramatic increase of the AQP5 mRNA and protein expression in the uterus at the 34th week of pregnancy, but they have not examined women in earlier periods. In this case, we can talk about a dramatic decrease of 36 weeks. It is not clear for which study group the results are presented in Fig.3. A similar observation should be made for AQP5 concentration in blood plasma (Fig.4). Is there any information available for the authors that reveals AQP5 levels in the uterus and blood during the second trimester?

Response 2:  We collected uterus samples (cc. 1cm3) at cesarean section. We could not collect samples before the 34th gestational week, because cesarean section before 34th gestational week is an urgent operation. The laboratory had to prepare for the sample reception, which is time-consuming. In addition, the laboratory is in a more remote building than the obstetrics operating room, and the sample transporter could not get there for the cesarean sections.

Reviewer#2 is right; we have corrected our statement about the 34th week in the manuscript (lines 225-228).

Figures 3 and 4 present the results of the mothers with normal weight. The text and figure legends were supplemented with this information.

Unfortunately, we have no information about the AQP5 levels in the uterus and blood during the second trimester. 

3.     Fig.5-7 compares AQP5 and TAC levels in women with normal weight and obesity, but the authors do not indicate when the study was conducted. The authors should indicate the age of pregnancy for these indicators. 

Response 3: The study period was between June 2023 and March 2024 (line 79). The ultrasound investigation is performed 16-48 hours before labor (line 106). The samples were collected from women with normal weight and obesity in their third trimester of pregnancy. We have corrected this in the article as well.

  1. The authors write that their data may be consistent with the report that obese mothers have larger placentas, than normal weight woman. It is important to include a reference to the report.

Response 4: It was one of the results of our study, as a new statement. You can see the data in Table 2.

  1. According to the authors, placental volumes are higher in pregnant women with obesity than normal weights, but all placental vascular indices show a slight decrease in cases complicated by obesity. Could it be due to increased water and placental edema?

Response 5: The placenta, a temporary organ responsible for transporting nutrition from the mother to the fetus, has been shown to influence birthweight, and its weight has been shown to be directly correlated with birthweight. The neonatal outcome will be the next article for us. We determined that obese pregnant women have more weighted neonates. The higher placental volume is the result of placental edema.

  • Risnes KR, Romundstad PR, Nilsen TI, Eskild A, Vatten LJ. Placental weight relative to birth weight and long-term cardiovascular mortality: findings from a cohort of 31,307 men and women. Am J Epidemiol 2009;170:622–31.
  • Hemachandra AH, Klebanoff MA, Duggan AK, Hardy JB, Furth SL. The association between intrauterine growth restriction in the full term infant and high blood pressure at age 7 years: results from the Collaborative Perinatal Project. Int J Epidemiol 2006;35:871–7.
  • Thame M, Osmond C, Wilks R, Bennett FI, Forrester TE. Second trimester placental volume and infant size at birth. Obstet Gynecol 2001;98:279–83.
  • Eskild A, Romundstad PR, Vatten LJ. Placental weight and birthweight: does the association differ between pregnancies with and without preeclampsia? Am J Obstet Gynecol 2009;201:595.e1–5.
  • Lao TT, Wong WM. Implications of a high placental ratio in pregnancies with appropriate-for-gestational age neonates. Gynecol Obstet Invest 2001;52:34–7.
  • Lao TT, Wong WM. Placental ratio – its relationship with mild maternal anaemia. Placenta 1997;18:593–6.
  • Strøm-Roum EM, Haavaldsen C, Tanbo TG, Eskild A. Placental weight relative to birthweight in pregnancies with maternal diabetes mellitus. Acta Obstet Gynecol Scand. 2013 Jul;92(7):783-9. doi: 10.1111/aogs.12104. Epub 2013 Mar 9.
  • Haavaldsen C, Samuelsen SO, Eskild A. Fetal death and placental weight/birthweight ratio: a population study. Acta Obstet Gynecol Scand. 2013 May;92(5):583-90. doi: 10.1111/aogs.12105.
  • Strøm-Roum EM, Haavaldsen C, Tanbo TG, Eskild A. Paternal age, placental weight and placental to birthweight ratio: a population-based study of 590,835 pregnancies. Hum Reprod. 2013 Nov;28(11):3126-33. doi: 10.1093/humrep/det299. Epub 2013 Jul 19.